# Gaussian Process Conditional Copulas with Applications to Financial Time Series

**José Miguel Hernández-Lobato**
Engineering Department
University of Cambridge
jmh233@cam.ac.uk

**James Robert Lloyd**
Engineering Department
University of Cambridge
jrl44@cam.ac.uk

**Daniel Hernández-Lobato**
Computer Science Department
Universidad Autónoma de Madrid
daniel.hernandez@uam.es

## Abstract

The estimation of dependencies between multiple variables is a central problem in the analysis of financial time series. A common approach is to express these dependencies in terms of a copula function. Typically the copula function is assumed to be constant but this may be inaccurate when there are covariates that could have a large influence on the dependence structure of the data. To account for this, a Bayesian framework for the estimation of conditional copulas is proposed. In this framework the parameters of a copula are non-linearly related to some arbitrary conditioning variables. We evaluate the ability of our method to predict time-varying dependencies on several equities and currencies and observe consistent performance gains compared to static copula models and other time-varying copula methods.

## 1 Introduction

Understanding dependencies within multivariate data is a central problem in the analysis of financial time series, underpinning common tasks such as portfolio construction and calculation of value-at-risk. Classical methods estimate these dependencies in terms of a covariance matrix (possibly time varying) which is induced from the data [4, 5, 7, 1]. However, a more general approach is to use copula functions to model dependencies [6]. Copulas have become popular since they separate the estimation of marginal distributions from the estimation of the dependence structure, which is completely determined by the copula.

The use of standard copula methods to estimate dependencies is likely to be inaccurate when the actual dependencies are strongly influenced by other covariates. For example, dependencies can vary with time or be affected by observations of other time series. Standard copula methods cannot handle such conditional dependencies. To address this limitation, we propose a probabilistic framework to estimate conditional copulas. Specifically we assume parametric copulas whose parameters are specified by unknown non-linear functions of arbitrary conditioning variables. These latent functions are approximated using Gaussian processes (GP) [17].

GPs have previously been used to model conditional copulas in [12] but that work only applies to copulas specified by a single parameter. We extend this work to accommodate copulas with multiple parameters. This is an important improvement since it allows the use of a richer set of copulas including Student's $t$ and asymmetric copulas. We demonstrate our method by choosing the conditioning variables to be time and evaluating its ability to estimate time-varying dependencies

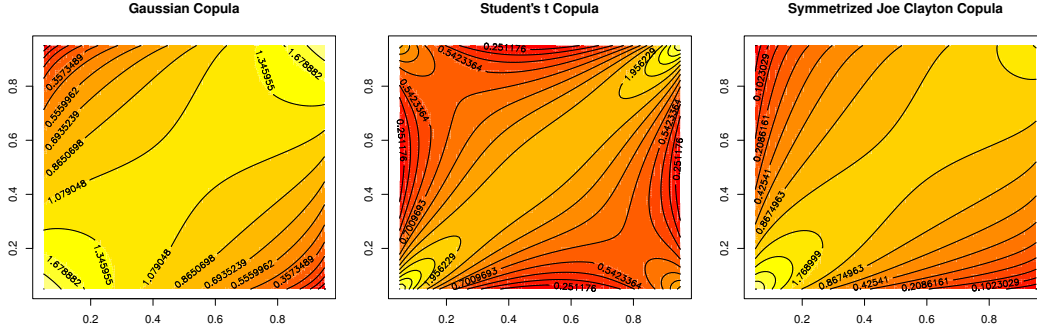

Figure 1: Left, Gaussian copula density for $\tau = 0.3$. Middle, Student's $t$ copula density for $\tau = 0.3$ and $\nu = 1$. Right, symmetrized Joe Clayton copula density for $\tau^U = 0.1$ and $\tau^L = 0.6$. The latter copula model is asymmetric along the main diagonal of the unit square.

on several currency and equity time series. Our method achieves consistently superior predictive performance compared to static copula models and other dynamic copula methods. These include models that allow their parameters to change with time, e.g. regime switching models [11] and methods proposing GARCH-style updates to copula parameters [20, 11].

## 2 Copulas and Conditional Copulas

Copulas provide a powerful framework for the construction of multivariate probabilistic models by separating the modeling of univariate marginal distributions from the modeling of dependencies between variables [6]. We focus on bivariate copulas since higher dimensional copulas are typically constructed using bivariate copulas as building blocks [e.g 2, 12].

Sklar's theorem [18] states that given two one-dimensional random variables, $X$ and $Y$, with continuous marginal cumulative distribution functions (cdfs) $F_X(X)$ and $F_Y(Y)$, we can express their joint cdf $F_{X,Y}$ as $F_{X,Y}(x,y) = C_{X,Y}[F_X(x), F_Y(y)]$, where $C_{X,Y}$ is the unique *copula* for $X$ and $Y$. Since $F_X(X)$ and $F_Y(Y)$ are marginally uniformly distributed on $[0,1]$, $C_{X,Y}$ is the cdf of a probability distribution on the unit square $[0,1] \times [0,1]$ with uniform marginals. Figure 1 shows plots of the copula densities for three parametric copula models: Gaussian, Student's $t$ and the symmetrized Joe Clayton (SJC) copulas. Copula models can be learnt in a two step process [10]. First, the marginals $F_X$ and $F_Y$ are learnt by fitting univariate models. Second, the data are mapped to the unit square by $U = F_X(X), V = F_Y(Y)$ (i.e. a probability integral transform) and then $C_{X,Y}$ is then fit to the transformed data.

### 2.1 Conditional Copulas

When one has access to a covariate vector $\mathbf{Z}$, one may wish to estimate a conditional version of a copula model i.e.

$$F_{X,Y|\mathbf{Z}}(x,y|\mathbf{z}) = C_{X,Y|\mathbf{Z}}\left[F_{X|\mathbf{Z}}(x|\mathbf{z}), F_{Y|\mathbf{Z}}(y|\mathbf{z})|\mathbf{z}\right] . \tag{1}$$

Here, the same two-step estimation process can be used to estimate $F_{X,Y|\mathbf{Z}}(x,y|\mathbf{z})$. The estimation of the marginals $F_{X|\mathbf{Z}}$ and $F_{Y|\mathbf{Z}}$ can be implemented using standard methods for univariate conditional distribution estimation. However, the estimation of $C_{X,Y|\mathbf{Z}}$ is constrained to have uniform marginal distributions; this is a problem that has only been considered recently [12]. We propose a general Bayesian non-parametric framework for the estimation of conditional copulas based on GPs and an alternating expectation propagation (EP) algorithm for efficient approximate inference.

## 3 Gaussian Process Conditional Copulas

Let $\mathcal{D}_{\mathbf{Z}} = \{\mathbf{z}_i\}_{i=1}^n$ and $\mathcal{D}_{U,V} = \{(u_i, v_i)\}_{i=1}^n$ where $(u_i, v_i)$ is a sample drawn from $C_{X,Y|\mathbf{z}_i}$. We assume that $C_{X,Y|\mathbf{Z}}$ is a parametric copula model $C_{\text{par}}[u, v|\theta_1(\mathbf{z}), \dots, \theta_k(\mathbf{z})]$ specified by $k$ parameters $\theta_1, \dots, \theta_k$ that may be functions of the conditioning variable $\mathbf{z}$. Let $\theta_i(\mathbf{z}) = \sigma_i[f_i(\mathbf{z})]$,

where $f_i$ is an arbitrary real function and $\sigma_i$ is a function that maps the real line to a set $\Theta_i$ of valid configurations for $\theta_i$. For example, $C_{\text{par}}$ could be a Student's $t$ copula. In this case, $k = 2$ and $\theta_1$ and $\theta_2$ are the correlation and the degrees of freedom in the Student's $t$ copula, $\Theta_1 = (-1, 1)$ and $\Theta_2 = (0, \infty)$. One could then choose $\sigma_1(\cdot) = 2\Phi(\cdot) - 1$, where $\Phi$ is the standard Gaussian cdf and $\sigma_2(\cdot) = \exp(\cdot)$ to satisfy the constraint sets $\Theta_1$ and $\Theta_2$ respectively.

Once we have specified the parametric form of $C_{\text{par}}$ and the mapping functions $\sigma_1, \ldots, \sigma_k$, we need to learn the latent functions $f_1, \ldots, f_k$. We perform a Bayesian non-parametric analysis by placing GP priors on these functions and computing their posterior distribution given the observed data.

Let $\mathbf{f}_i = (f_i(\mathbf{z}_1), \ldots, f_i(\mathbf{z}_n))^{\text{T}}$. The prior distribution for $\mathbf{f}_i$ given $\mathcal{D}_{\mathbf{Z}}$ is $p(\mathbf{f}_i|\mathcal{D}_{\mathbf{Z}}) = \mathcal{N}(\mathbf{f}_i|\mathbf{m}_i, \mathbf{K}_i)$, where $\mathbf{m}_i = (m_i(\mathbf{z}_1), \ldots, m_i(\mathbf{z}_n))^{\text{T}}$ for some mean function $m_i(\mathbf{z})$ and $\mathbf{K}_i$ is an $n \times n$ covariance matrix generated by the squared exponential covariance function, i.e.

$$[\mathbf{K}_i]_{jk} = \text{Cov}[f_i(\mathbf{z}_j), f_i(\mathbf{z}_k)] = \beta_i \exp\left\{-(\mathbf{z}_j - \mathbf{z}_k)^{\text{T}}\text{diag}(\boldsymbol{\lambda}_i)(\mathbf{z}_j - \mathbf{z}_k)\right\} + \gamma_i \,, \qquad (2)$$

where $\boldsymbol{\lambda}_i$ is a vector of inverse length-scales and $\beta_i$, $\gamma_i$ are amplitude and noise parameters. The posterior distribution for $\mathbf{f}_1, \ldots, \mathbf{f}_k$ given $\mathcal{D}_{U,V}$ and $\mathcal{D}_{\mathbf{Z}}$ is

$$p(\mathbf{f}_1, \ldots, \mathbf{f}_k|\mathcal{D}_{U,V}, \mathcal{D}_{\mathbf{Z}}) = \frac{\left[\prod_{i=1}^n c_{\text{par}}\big[u_i, v_i|\sigma_1\left[f_1(\mathbf{z}_i)\right], \ldots, \sigma_k\left[f_k(\mathbf{z}_i)\right]\big]\right]\left[\prod_{i=1}^k \mathcal{N}(\mathbf{f}_i|\mathbf{m}_i, \mathbf{K}_i)\right]}{p(\mathcal{D}_{U,V}|\mathcal{D}_{\mathbf{Z}})} \,, \quad (3)$$

where $c_{\text{par}}$ is the density of the parametric copula model and $p(\mathcal{D}_{U,V}|\mathcal{D}_{\mathbf{Z}})$ is a normalization constant often called the *model evidence*. Given a particular value of $\mathbf{Z}$ denoted by $\mathbf{z}^\star$, we can make predictions about the conditional distribution of $U$ and $V$ using the standard GP prediction formula

$$p(u^\star, v^\star|\mathbf{z}^\star) = \int c_{\text{par}}(u^\star, v^\star|\sigma_1[f_1^\star], \ldots, \sigma_k[f_k^\star])p(\mathbf{f}^\star|\mathbf{f}_1, \ldots, \mathbf{f}_k, \mathbf{z}^\star, \mathcal{D}_{\mathbf{Z}})$$

$$p(\mathbf{f}_1, \ldots, \mathbf{f}_k|\mathcal{D}_{U,V}, \mathcal{D}_{\mathbf{Z}})\, d\mathbf{f}_1 \cdots d\mathbf{f}_k\, d\mathbf{f}^\star \,, \qquad (4)$$

where $\mathbf{f}^\star = (f_1^\star, \ldots, f_k^\star)^{\text{T}}$, $p(\mathbf{f}^\star|\mathbf{f}_1, \ldots, \mathbf{f}_k, \mathbf{z}^\star, \mathcal{D}_{\mathbf{Z}}) = \prod_{i=1}^k p(f_i^\star|\mathbf{f}_i, \mathbf{z}^\star, \mathcal{D}_{\mathbf{Z}})$, $f_i^\star = f_i(\mathbf{z}^\star)$, $p(f_i^\star|\mathbf{f}_i, \mathbf{z}^\star, \mathcal{D}_{\mathbf{Z}}) = \mathcal{N}(f_i^\star|m_i(\mathbf{z}^\star) + \mathbf{k}_i^{\text{T}}\mathbf{K}_i^{-1}(\mathbf{f}_i - \mathbf{m}_i), k_i - \mathbf{k}_i^{\text{T}}\mathbf{K}_i^{-1}\mathbf{k}_i)$, $k_i = \text{Cov}[f_i(\mathbf{z}^\star), f_i(\mathbf{z}^\star)]$ and $\mathbf{k}_i = (\text{Cov}[f_i(\mathbf{z}^\star), f_i(\mathbf{z}_1)], \ldots, \text{Cov}[f_i(\mathbf{z}^\star), f_i(\mathbf{z}_n)])^{\text{T}}$. Unfortunately, (3) and (4) cannot be computed analytically, so we approximate them using expectation propagation (EP) [13].

### 3.1 An Alternating EP Algorithm for Approximate Bayesian Inference

The joint distribution for $\mathbf{f}_1, \ldots, \mathbf{f}_k$ and $\mathcal{D}_{U,V}$ given $\mathcal{D}_{\mathbf{Z}}$ can be written as a product of $n + k$ factors:

$$p(\mathbf{f}_1, \ldots, \mathbf{f}_k, \mathcal{D}_{U,V}|\mathcal{D}_{\mathbf{Z}}) = \left[\prod_{i=1}^n g_i(f_{1i}, \ldots, f_{ki},)\right]\left[\prod_{i=1}^k h_i(\mathbf{f}_i)\right] \,, \qquad (5)$$

where $f_{ji} = f_j(\mathbf{z}_i)$, $h_i(\mathbf{f}_i) = \mathcal{N}(\mathbf{f}_i|\mathbf{m}_i, \mathbf{K}_i)$ and $g_i(f_{1i}, \ldots, f_{ki}) = c_{\text{par}}[u_i, v_i|\sigma_1[f_{1i}], \ldots, \sigma_k[f_{ki}]]$. EP approximates each factor $g_i$ with an approximate Gaussian factor $\tilde{g}_i$ that may not integrate to one, i.e. $\tilde{g}_i(f_{1i}, \ldots, f_{ki}) = s_i \prod_{j=1}^k \exp\left\{-(f_{ji} - \tilde{m}_{ji})^2/[2\tilde{v}_{ji}]\right\}$, where $s_i > 0$, $\tilde{m}_{ji}$ and $\tilde{v}_{ji}$ are parameters to be calculated by EP. The other factors $h_i$ already have a Gaussian form so they do not need to be approximated. Since all the $\tilde{g}_i$ and $h_i$ are Gaussian, their product is, up to a normalization constant, a multivariate Gaussian distribution $q(\mathbf{f}_1, \ldots, \mathbf{f}_k)$ which approximates the exact posterior (3) and factorizes across $\mathbf{f}_1, \ldots, \mathbf{f}_k$. The predictive distribution (4) is approximated by first integrating $p(\mathbf{f}^\star|\mathbf{f}_1, \ldots, \mathbf{f}_k, \mathbf{z}^\star, \mathcal{D}_{\mathbf{Z}})$ with respect to $q(\mathbf{f}_1, \ldots, \mathbf{f}_k)$. This results in a factorized Gaussian distribution $q^\star(\mathbf{f}^\star)$ which approximates $p(\mathbf{f}^\star|\mathcal{D}_{U,V}, \mathcal{D}_{\mathbf{Z}})$. Finally, (4) is approximated by Monte-Carlo by sampling from $q^\star$ and then averaging $c_{\text{par}}(u^\star, v^\star|\sigma_1[f_1^\star], \ldots, \sigma_k[f_k^\star])$ over the samples.

EP iteratively updates each $\tilde{g}_i$ until convergence by first computing $q^{\backslash i} \propto q/\tilde{g}_i$ and then minimizing the Kullback-Leibler divergence [3] between $g_i q^{\backslash i}$ and $\tilde{g}_i q^{\backslash i}$. This involves updating $\tilde{g}_i$ so that the first and second marginal moments of $g_i q^{\backslash i}$ and $\tilde{g}_i q^{\backslash i}$ match. However, it is not possible to compute the moments of $g_i q^{\backslash i}$ analytically due to the complicated form of $g_i$. A solution is to use numerical methods to compute these $k$-dimensional integrals. However, this typically has an exponentially large computational cost in $k$ which is prohibitive for $k > 1$. Instead we perform an additional approximation when computing the marginal moments of $f_{ji}$ with respect to $g_i q^{\backslash i}$. Without loss of

generality, assume that we want to compute the expectation of $f_{1i}$ with respect to $g_i q^{\setminus i}$. We make the following approximation:

$$\int f_{1i} g_i(f_{1i}, \ldots, f_{ki}) q^{\setminus i}(f_{1i}, \ldots, f_{ki}) \, df_{1i}, \ldots, df_{ki} \approx$$

$$C \times \int f_{1i} g_i(f_{1i}, \bar{f}_{2i}, \ldots, \bar{f}_{ki}) q^{\setminus i}(f_{1i}, \bar{f}_{2i}, \ldots, \bar{f}_{ki}) \, df_{1i}, \quad (6)$$

where $\bar{f}_{1i}, \ldots, \bar{f}_{ki}$ are the means of $f_{1i}, \ldots, f_{ki}$ with respect to $q^{\setminus i}$, and $C$ is a constant that approximates the width of the integrand around its maximum in all dimensions except $f_{1i}$. In practice all moments are normalized by the 0-th moment so $C$ can be ignored. The right hand side of (6) is a one-dimensional integral that can be easily computed using numerical techniques. The approximation above is similar to approximating an integral by the product of the maximum value of the integrand and an estimate of its width. However, instead of maximizing $g_i(f_{1i}, \ldots, f_{ki}) q^{\setminus i}(f_{1i}, \ldots, f_{ki})$ with respect to $f_{2i}, \ldots, f_{ki}$, we are maximizing $q^{\setminus i}$. This is a much easier task because $q^{\setminus i}$ is Gaussian and its maximizer is its own mean vector. In practice, $g_i(f_{1i}, \ldots, f_{ki})$ is very flat when compared to $q^{\setminus i}$ and the maximizer of $q^{\setminus i}$ approximates well the maximizer of $g_i(f_{1i}, \ldots, f_{ki}) q^{\setminus i}(f_{1i}, \ldots, f_{ki})$.

Since $q$ factorizes across $\mathbf{f}_1, \ldots, \mathbf{f}_k$ (as well as $q^{\setminus i}$), our implementation of EP decouples into $k$ EP sub-routines among which we alternate; the $j$-th sub-routine approximates the posterior distribution of $\mathbf{f}_j$ using as input the means of $q^{\setminus i}$ generated by the other EP sub-routines. Each sub-routine finds a Gaussian approximation to a set of $n$ one-dimensional factors; one factor per data point. In the $j$-th EP sub-routine, the $i$-th factor is given by $g_i(f_{1i}, \ldots, f_{ki})$, where each $\{f_{1i}, \ldots, f_{ki}\} \setminus \{f_{ji}\}$ is kept fixed to the current mean of $q^{\setminus i}$, as estimated by the other EP sub-routines. We iteratively alternate between sub-routines, running each one until convergence before re-running the next one. Convergence is achieved very quickly; we only run each EP sub-routine four times.

The EP sub-routines are implemented using the parallel EP update scheme described in [21]. To speed up GP related computations, we use the generalized FITC approximation [19, 14]: Each $n \times n$ covariance matrix $\mathbf{K}_i$ is approximated by $\mathbf{K}_i' = \mathbf{Q}_i + \operatorname{diag}(\mathbf{K}_i - \mathbf{Q}_i)$, where $\mathbf{Q}_i = \mathbf{K}_{nn_0}^i [\mathbf{K}_{n_0 n_0}^i]^{-1} [\mathbf{K}_{nn_0}^i]^{\mathrm{T}}$, $\mathbf{K}_{n_0 n_0}^i$ is the $n_0 \times n_0$ covariance matrix generated by evaluating (2) at $n_0 \ll n$ *pseudo-inputs*, and $\mathbf{K}_{nn_0}^i$ is the $n \times n_0$ matrix with the covariances between training points and pseudo-inputs. The cost of EP is $O(knn_0^2)$. Each time we call the $j$-th EP subroutine, we optimize the corresponding kernel hyper-parameters $\boldsymbol{\lambda}_j$, $\beta_j$ and $\gamma_j$ and the pseudo-inputs by maximizing the EP approximation of the model evidence [17].

# 4 Related Work

The model proposed here is an extension of the conditional copula model of [12]. In the case of bivariate data and a copula based on one parameter the models are identical. We have extended the approximate inference for this model to accommodate copulas with multiple parameters; previously computationally infeasible due to requiring the numerical calculation of multidimensional integrals within an inner loop of EP inference. We have also demonstrated that one can use this model to produce excellent predictive results on financial time series by conditioning the copula on time.

## 4.1 Dynamic Copula Models

In [11] a dynamic copula model is proposed based on a two-state hidden Markov model (HMM) ($S_t \in \{0, 1\}$) that assumes that the data generating process changes between two regimes of low/high correlation. At any time $t$ the copula density is Student's $t$ with different parameters for the two values of the hidden state $S_t$. Maximum likelihood estimation of the copula parameters and transition probabilities is performed using an EM algorithm [e.g. 3].

A time-varying correlation (TVC) model based on the Student's $t$ copula is described in [20, 11]. The correlation parameter[1] of a Student's $t$ copula is assumed to satisfy $\rho_t = (1 - \alpha - \beta)\rho + \alpha\varepsilon_{t-1} + \beta\rho_{t-1}$, where $\varepsilon_{t-1}$ is the empirical correlation of the previous 10 observations and $\rho$, $\alpha$ and $\beta$ satisfy $-1 \leq \rho \leq 1$, $0 \leq \alpha, \beta \leq 1$ and $\alpha + \beta \leq 1$. The number of degrees of freedom $\nu$

is assumed to be constant. The previous formula is the GARCH equation for correlation instead of variance. Estimation of $\rho$, $\alpha$, $\beta$ and $\nu$ is easily performed by maximum likelihood.

In [15] a dynamic copula based on the SJC copula (DSJCC) is introduced. In this method, the parameters $\tau^U$ and $\tau^L$ of an SJC copula are assumed to depend on time according to

$$\tau^U(t) = 0.01 + 0.98\Lambda\left[\omega_U + \alpha_U\varepsilon_{t-1} + \beta_U\tau^U(t-1)\right], \tag{7}$$

$$\tau^L(t) = 0.01 + 0.98\Lambda\left[\omega_L + \alpha_L\varepsilon_{t-1} + \beta_L\tau^L(t-1)\right], \tag{8}$$

where $\Lambda[\cdot]$ is the logistic function, $\varepsilon_{t-1} = \frac{1}{10}\sum_{j=1}^{10}|u_{t-j} - v_{t-j}|$, $(u_t, v_t)$ is a copula sample at time $t$ and the constants are used to avoid numerical instabilities. These formulae are the GARCH equation for correlations, with an additional logistic function to constrain parameter values. The estimation of $\omega_U, \alpha_U, \beta_U, \omega_L, \alpha_L$ and $\beta_L$ is performed by maximum likelihood.

We go beyond this prior work by allowing copula parameters to depend on an arbitrary conditioning variables rather than time alone. Also, the models above either assume Markov independence or GARCH-like updates to copula parameters. These assumptions have been empirically proven to be effective for the estimation of univariate variances, but the consistent performance gains of our proposed method suggest these assumptions are less applicable for the estimation of dependencies.

## 4.2 Other Dynamic Covariance Models

A direct extension of the GARCH equations to multiple time series, VEC, was proposed by [5]. Let $\mathbf{x}(t)$ be a multivariate time series assumed to satisfy $\mathbf{x}(t) \sim \mathcal{N}(0, \Sigma(t))$. VEC$(p, q)$ models the dynamics of $\Sigma(t)$ by an equation of the form

$$\text{vech}(\Sigma(t)) = c + \sum_{k=1}^{p} A_k \text{vech}(\mathbf{x}(t-k)\mathbf{x}(t-k)^\mathrm{T}) + \sum_{k=1}^{q} B_k \text{vech}(\Sigma(t-k)) \tag{9}$$

where vech is the operation that stacks the lower triangular part on a matrix into a column vector. The VEC model has a very large number of parameters and hence a more commonly used model is the BEKK$(p, q)$ model [7] which assumes the following dynamics

$$\Sigma(t) = C^\mathrm{T}C + \sum_{k=1}^{p} A_k^\mathrm{T}\mathbf{x}(t-k)\mathbf{x}(t-k)^\mathrm{T}A_k + \sum_{k=1}^{q} B_k^\mathrm{T}\Sigma(t-k)B_k. \tag{10}$$

This model also has many parameters and many restricted versions of these models have been proposed to avoid over-fitting (see e.g. section 2 of [1]).

An alternative solution to over-fitting due to over-parameterization is the Bayesian approach of [23] where Bayesian inference is performed in a dynamic BEKK$(1, 1)$ model. Other Bayesian approaches include the non-parametric generalized Wishart process [22, 8]. In these works $\Sigma(t)$ is modeled by a generalized Wishart process i.e.

$$\Sigma(t) = \sum_{i=1}^{\nu} L\mathbf{u}_i(t)\mathbf{u}_i(t)^\mathrm{T}L^\mathrm{T} \tag{11}$$

where $u_{id}(\cdot)$ are distributed as independent GPs.

## 5 Experiments

We evaluate the proposed Gaussian process conditional copula models (GPCC) on a one-step-ahead prediction task with synthetic data and financial time series. We use time as the conditioning variable and consider three parametric copula families; Gaussian (GPCC-G), Student's $t$ (GPCC-T) and symmetrized Joe Clayton (GPCC-SJC). The parameters of these copulas are presented in Table 1 along with the transformations used to model them. Figure 1 shows plots of the densities of these three parametric copula models. The code and data are publicly available at http://jmhl.org.

| Copula | Parameters | Transformation | Synthetic parameter function |
|---|---|---|---|
| Gaussian | correlation, $\tau$ | $0.99(2\Phi[f(t)] - 1)$ | $\tau(t) = 0.3 + 0.2\cos(t\pi/125)$ |
| Student's $t$ | correlation, $\tau$ | $0.99(2\Phi[f(t)] - 1)$ | $\tau(t) = 0.3 + 0.2\cos(t\pi/125)$ |
| | degrees of freedom, $\nu$ | $1 + 10^6\Phi[g(t)]$ | $\nu(t) = 1 + 2(1 + \cos(t\pi/250))$ |
| SJC | upper dependence, $\tau^U$ | $0.01 + 0.98\Phi[g(t)]$ | $\tau^U(t) = 0.1 + 0.3(1 + \cos(t\pi/125))$ |
| | lower dependence, $\tau^L$ | $0.01 + 0.98\Phi[g(t)]$ | $\tau^L(t) = 0.1 + 0.3(1 + \cos(t\pi/125 + \pi/2))$ |

Table 1: Copula parameters, modeling formulae and parameter functions used to generate synthetic data. $\Phi$ is the standard Gaussian cumulative density function $f$ and $g$ are GPs.

The three variants of GPCC were compared against three dynamic copula methods and three constant copula models. The three dynamic methods include the HMM based model, TVC and DSJCC introduced in Section 4. The three constant copula models use Gaussian, Student's $t$ and SJC copulas with parameter values that do not change with time (CONST-G, CONST-T and CONST-SJC). We perform a one-step-ahead rolling-window prediction task on bivariate time series $\{(u_t, v_t)\}$. Each model is trained on the first $n_W$ data points and the predictive log-likelihood of the $(n_W+1)-$th data point is recorded, where $n_W = 1000$. This is then repeated, shifting the training and test windows forward by one data point. The methods are then compared by average predictive log-likelihood; an appropriate performance measure for copula estimation since copulas are probability distributions.

## 5.1 Synthetic Data

We generated three synthetic datasets of length 5001 from copula models (Gaussian, Student's $t$, SJC) whose parameters vary as periodic functions of time, as specified in Table 1. Table 2 reports the average predictive log-likelihood for each method on each synthetic time series. The results of the best performing method on each synthetic time series are shown in bold. The results of any other method are underlined when the differences with respect to the best performing method are not statistically significant according to a paired $t$ test at $\alpha = 0.05$.

GPCC-T and GPCC-SJC obtain the best results in the Student's $t$ and SJC time series respectively. However, HMM is the best performing method for the Gaussian time series. This technique successfully captures the two regimes of low/high correlation corresponding to the peaks and troughs of the sinusoid that maps time $t$ to correlation $\tau$. The proposed methods GPCC-[G,T,SJC] are more flexible and hence less efficient than HMM in this particular problem. However, HMM performs significantly worse in the Student's $t$ and SJC time series since the different periods for the different copula parameter functions cannot be captured by a two state model. Figure 2 shows how GPCC-T successfully tracks $\tau(t)$ and $\nu(t)$ in the Student's $t$ time series. The plots display the mean (red) and confidence bands (orange, 0.1 and 0.9 quantiles) for the predictive distribution of $\tau(t)$ and $\nu(t)$ as well as the ground truth values (blue). Finally, Table 2 also shows that the static copula methods CONST-[G,T,SJC] are usually outperformed by all dynamic techniques GPCC-[G,T,SJC], DSJCC, TVC and HMM.

## 5.2 Foreign Exchange Time Series

We evaluated each method on the daily logarithmic returns of nine currencies shown in Table 3 (all priced with respect to the U.S. dollar).The date range of the data is 02-01-1990 to 15-01-2013; a total of 6011 observations. We evaluated the methods on eight bivariate time series, pairing each currency pair with the Swiss franc (CHF). CHF is known to be a *safe haven* currency, meaning that investors flock to it during times of uncertainty [16]. Consequently we expect correlations between CHF and other currencies to have large variability across time in response to changes in financial conditions.

We first process our data using an asymmetric AR(1)-GARCH(1,1) process with non-parametric innovations [9] to estimate the univariate marginal cdfs at all time points. We train this GARCH model on $n_W = 2016$ data points and then predict the cdf of the next data point; subsequent cdfs are predicted by shifting the training window by one data point in a rolling-window methodology. The cdf estimates are used to transform the raw logarithmic returns $(x_t, y_t)$ into a pseudo-sample of the underlying copula $(u_t, v_t)$ as described in Section 2. We note that any method for predicting univariate cdfs could have been used to produce pseudo-samples from the copula. We then perform

the rolling-window predictive likelihood experiment on the transformed data. The results are shown in Table 4; overall the best technique is GPCC-T, followed by GPCC-G. The dynamic copula methods GPCC-[G,T,SJC], HMM, and TVC outperform the static methods CONST-[G,T,SJC] in all the analyzed series. The dynamic method DSJCC occasionally performed poorly; worse than the static methods for 3 experiments.

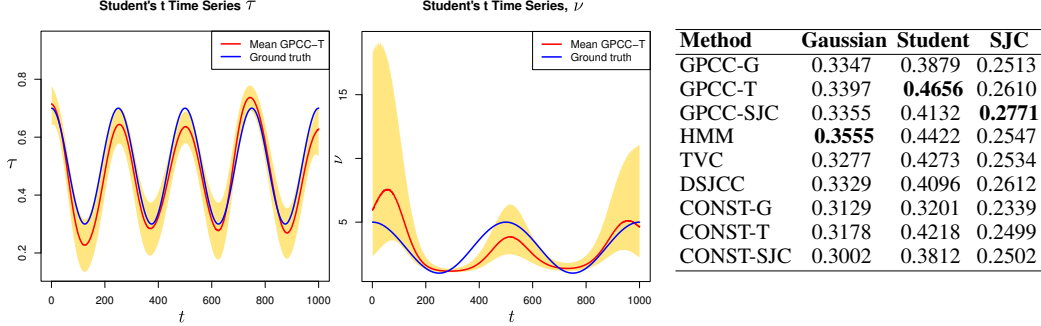

| Method | Gaussian | Student | SJC |
|---|---|---|---|
| GPCC-G | 0.3347 | 0.3879 | 0.2513 |
| GPCC-T | 0.3397 | **0.4656** | 0.2610 |
| GPCC-SJC | 0.3355 | 0.4132 | **0.2771** |
| HMM | **0.3555** | 0.4422 | 0.2547 |
| TVC | 0.3277 | 0.4273 | 0.2534 |
| DSJCC | 0.3329 | 0.4096 | 0.2612 |
| CONST-G | 0.3129 | 0.3201 | 0.2339 |
| CONST-T | 0.3178 | 0.4218 | 0.2499 |
| CONST-SJC | 0.3002 | 0.3812 | 0.2502 |

Figure 2: Predictions made by GPCC-T for $\nu(t)$ and $\tau(t)$ on the synthetic time series sampled from a Student's $t$ copula.

Table 2: Avg. test log-likelihood of each method on each time series.

| Code | Currency Name |
|---|---|
| CHF | Swiss Franc |
| AUD | Australian Dollar |
| CAD | Canadian Dollar |
| JPY | Japanese Yen |
| NOK | Norwegian Krone |
| SEK | Swedish Krona |
| EUR | Euro |
| NZD | New Zeland Dollar |
| GBP | British Pound |

| Method | AUD | CAD | JPY | NOK | SEK | EUR | GBP | NZD |
|---|---|---|---|---|---|---|---|---|
| GPCC-G | 0.1260 | 0.0562 | **0.1221** | 0.4106 | 0.4132 | 0.8842 | 0.2487 | 0.1045 |
| GPCC-T | **0.1319** | **0.0589** | 0.1201 | **0.4161** | **0.4192** | **0.8995** | **0.2514** | **0.1079** |
| GPCC-SJC | 0.1168 | 0.0469 | 0.1064 | 0.3941 | 0.3905 | 0.8287 | 0.2404 | 0.0921 |
| HMM | 0.1164 | 0.0478 | 0.1009 | 0.4069 | 0.3955 | 0.8700 | 0.2374 | 0.0926 |
| TVC | 0.1181 | 0.0524 | 0.1038 | 0.3930 | 0.3878 | 0.7855 | 0.2301 | 0.0974 |
| DSJCC | 0.0798 | 0.0259 | 0.0891 | 0.3994 | 0.3937 | 0.8335 | 0.2320 | 0.0560 |
| CONST-G | 0.0925 | 0.0398 | 0.0771 | 0.3413 | 0.3426 | 0.6803 | 0.2085 | 0.0745 |
| CONST-T | 0.1078 | 0.0463 | 0.0898 | 0.3765 | 0.3760 | 0.7732 | 0.2231 | 0.0875 |
| CONST-SJC | 0.1000 | 0.0425 | 0.0852 | 0.3536 | 0.3544 | 0.7113 | 0.2165 | 0.0796 |

Table 3: Currencies.

Table 4: Avg. test log-likelihood of each method on the currency data.

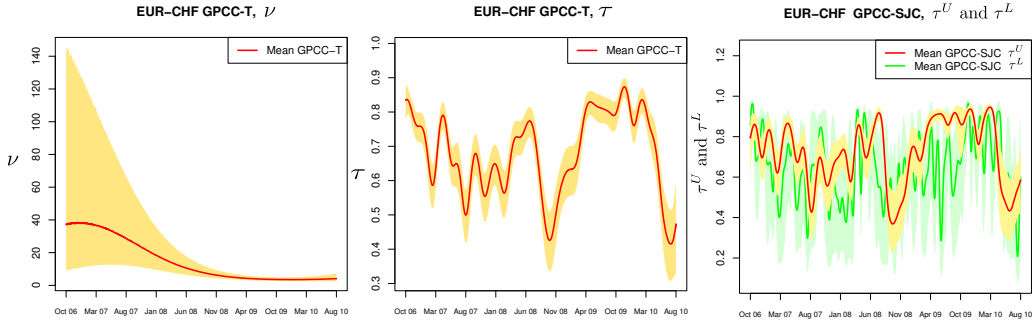

Figure 3: Left and middle, predictions made by GPCC-T for $\nu(t)$ and $\tau(t)$ on the time series EUR-CHF when trained on data from 10-10-2006 to 09-08-2010. There is a significant reduction in $\nu(t)$ at the onset of the 2008-2012 global recession. Right, predictions made by GPCC-SJC for $\tau^U(t)$ and $\tau^L(t)$ when trained on the same time-series data. The predictions for $\tau^L(t)$ are much more erratic than those for $\tau^U(t)$.

The proposed method GPCC-T can capture changes across time in the parameters of the Student's $t$ copula. The left and middle plots in Figure 3 show predictions for $\nu(t)$ and $\tau(t)$ generated by GPCC-T. In the left plot, we observe a reduction in $\nu(t)$ at the onset of the 2008-2012 global recession indicating that the return series became more prone to outliers. The plot for $\tau(t)$ (middle) also shows large changes across time. In particular, we observe large drops in the dependence level between EUR-USD and CHF-USD during the fall of 2008 (at the onset of the global recession) and the summer of 2010 (corresponding to the worsening European sovereign debt crisis).

For comparison, we include predictions for $\tau^L(t)$ and $\tau^U(t)$ made by GPCC-SJC in the right plot of Figure 3. In this case, the prediction for $\tau^U(t)$ is similar to the one made by GPCC-T for $\tau(t)$,

but the prediction for $\tau^L(t)$ is much noisier and erratic. This suggests that GPCC-SJC is less robust than GPCC-T. All the copula densities in Figure 1 take large values in the proximity of the points (0,0) and (1,1) i.e. positive correlation. However, the Student's $t$ copula is the only one of these three copulas which can take high values in the proximity of the points (0,1) and (1,0) i.e. negative correlation. The plot in the left of Figure 3 shows how $\nu(t)$ takes very low values at the end of the time period, increasing the robustness of GPCC-T to negatively correlated outliers.

## 5.3 Equity Time Series

As a further comparison, we evaluated each method on the logarithmic returns of 8 equity pairs, from the same date range and processed using the same AR(1)-GARCH(1,1) model discussed previously. The equities were chosen to include pairs with both high correlation (e.g. RBS and BARC) and low correlation (e.g. AXP and BA).

The results are shown in Table 5; again the best technique is GPCC-T, followed by GPCC-G.

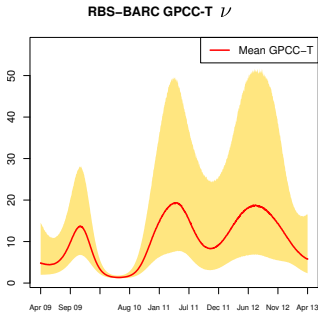

| Method | HD HON | AXP BA | CNW CSX | ED EIX | HPQ IBM | BARC HSBC | RBS BARC | RBS HSBC |
|---|---|---|---|---|---|---|---|---|
| GPCC-G | 0.1247 | 0.1133 | 0.1450 | **0.2072** | 0.1536 | 0.2424 | 0.3401 | 0.1860 |
| GPCC-T | **0.1289** | **0.1187** | **0.1499** | 0.2059 | **0.1591** | 0.2486 | **0.3501** | **0.1882** |
| GPCC-SJC | 0.1210 | 0.1095 | 0.1399 | 0.1935 | 0.1462 | 0.2342 | 0.3234 | 0.1753 |
| HMM | 0.1260 | 0.1119 | 0.1458 | 0.2040 | 0.1511 | **0.2486** | 0.3414 | 0.1818 |
| TVC | 0.1251 | 0.1119 | 0.1459 | 0.2011 | 0.1511 | 0.2449 | 0.3336 | 0.1823 |
| DSJCC | 0.0935 | 0.0750 | 0.1196 | 0.1721 | 0.1163 | 0.2188 | 0.3051 | 0.1582 |
| CONST-G | 0.1162 | 0.1027 | 0.1288 | 0.1962 | 0.1325 | 0.2307 | 0.2979 | 0.1663 |
| CONST-T | 0.1239 | 0.1091 | 0.1408 | 0.2007 | 0.1481 | 0.2426 | 0.3301 | 0.1775 |
| CONST-SJC | 0.1175 | 0.1046 | 0.1307 | 0.1891 | 0.1373 | 0.2268 | 0.2992 | 0.1639 |

Figure 4: Prediction for $\nu(t)$ on RBS-BARC.   Table 5: Average test log-likelihood for each method on each pair of stocks.

Figure 4 shows predictions for $\nu(t)$ generated by GPCC-T. We observe low values of $\nu$ during 2010 suggesting that a Gaussian copula would be a bad fit to the data. Indeed, GPCC-G performs significantly worse than GPCC-T on this equity pair.

## 6 Conclusions and Future Work

We have proposed an inference scheme to fit a conditional copula model to multivariate data where the copula is specified by multiple parameters. The copula parameters are modeled as unknown non-linear functions of arbitrary conditioning variables. We evaluated this framework by estimating time-varying copula parameters for bivariate financial time series. Our method consistently outperforms static copula models and other dynamic copula models.

In this initial investigation we have focused on bivariate copulas. Higher dimensional copulas are typically constructed using bivariate copulas as building blocks [2, 12]. Our framework could be applied to these constructions and our empirical predictive performance gains will likely transfer to this setting. Evaluating the effectiveness of this approach compared to other models of multivariate covariance would be a profitable area of empirical research.

One could also extend the analysis presented here by including additional conditioning variables as well as time. For example, including a prediction of univariate volatility as a conditioning variable would allow copula parameters to change in response to changing volatility. This would pose inference challenges as the dimension of the GP increases, but could create richer models.

### Acknowledgements

We thank David López-Paz and Andrew Gordon Wilson for interesting discussions. José Miguel Hernández-Lobato acknowledges support from Infosys Labs, Infosys Limited. Daniel Hernandez-Lobato acknowledges support from the Spanish Dirección General de Investigación, project ALLS (TIN2010-21575-C02-02).

## Footnotes

[1]The parameterization used in this paper is related by $\rho = \sin(0.5\tau\pi)$

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
