[Reviews · NeurIPS 2013]

Submitted by Assigned_Reviewer_4

Paper summary:

The authors propose a Bayesian inference procedure for conditional bivariate copula models, where the copula models can have multiple parameters. The copula parameters are related to conditioning variables by a composition of parameter mapping functions and arbitrary latent functions with Gaussian process priors. The inference procedure for this model is based on expectation propagation. The method is evaluated on synthetic data and on financial time series data and compared to alternative copula-based models. The paper also includes an overview of related methods.


Quality:

The proposed method is sound. The authors include a description of related methods and list strengths and weaknesses of their method.


Clarity:

The paper is written clearly and concisely. It makes a very mature impression. I could not find any inconsistencies or typos. The figures and tables complement the text seamlessly, equations are clear and well explained and the text is flawless.


Originality:

The model and its inference procedure is an extension of a similar conditional copula model that was restricted to copulas with only one parameter. The present work generalizes this approach to allow more than one copula parameter. To my knowledge, this extension is novel.


Significance:

The flexibility of copula families with only one parameter is limited and the applicability of more flexible families is a huge advantage. Performance gains compared to the alternative methods are remarkable. This result is very promising.
Summary: The paper extends an inference procedure for conditional copula models. The contribution is important and well done.

Submitted by Assigned_Reviewer_6

This paper presents an interesting approach for modeling dependences between variables based on Gaussian process conditional copulas. It employs Gaussian process priors on the latent functions to model their interactions. Then an alternating EP algorithm is proposed for the approximate Bayesian inference. Experimental results on both synthetic data and two realworld data indicate the better performance of conditional copula models, especially the one based on student t copula.

In general, this paper is well written and perhaps useful for dependence modeling in time series data. The EP inference is also implementable for the posterior distribution in the paper. A major problem of this paper is that some details are missing:
(1) How the parameters in the copula model (e.g., \alpha, \beta, \omega for the Symmetrized Joe Clayton Copula) in are calculated? I assume this paper sets them to constants. How to choose them for a given time series data?
(2) The details of the parameters the exponential covariance function in Gaussian process are missing.
(3) How to choose the number of latent functions k?


Summary: This paper discusses an interesting topic and derives an implementable inference method. However, some details on the parameter settings are missing.

Submitted by Assigned_Reviewer_7

The paper tackles the problem of fitting conditional copula models. The paper goes beyond typical constant copula functions and aim to model cases where parameters of the copula are functions of other (time) varying aspects of the problem.

The core solution is essentially modeling the parameters space as Gaussian Processes. Starting from a Gaussian Process prior, the information from observed data are incorporated via Expectation Propagation for approximate inference of the posterior distribution.

The paper is extremely well written and easy to follow. The problem is well motivated and most importantly illustrations with HMM based and multivariate time series copulas are very nicely written. The experiments are sufficient as well.
Summary: Overall, this is a nice paper that motivates a novel problem and provides a reasonable solution with excellent experimental illustrations.
Author Feedback

Author rebuttal: We would like to thank all of the reviewers for their time and comments. We are also very pleased to receive unanimously positive feedback. Below, we answer three specific questions and correct an error in our submission (which fortunately only strengthens our conclusions).


Reviewer 6 asked some specific questions which we answer below - we will update the manuscript to make these points clearer.

1 ) The parameters alpha, beta and omega in the DSJCC method are found by maximum likelihood.


2 ) The covariance function for the Gaussian processes is the squared exponential kernel (for the parametric form see equation (2) or e.g. Rasmussen and Williams). All parameters of the kernels were found by maximizing the EP approximation to the model evidence.


3 ) The number of latent functions k is determined by the number of parameters of the parametric copula model. There is one latent function for each parameter.

To all reviewers:

We would also like to correct a “copy-paste” error we made just before submission. The numbers in table 4 were mistakenly set to be identical to those in table 5. Fortunately, the discussion in the manuscript was based on the correct figures and the correct values only strengthen our conclusions. We include the correct figures for Table 4 below. The pattern of bold and underlined numbers is similar to that in Table 5, which explains why we did not spot the error in our submission.

We include below the average test log-likelihood for each method, GPCC-G, GPCC-T, GPCC-SJC, HMM, TVC, DSJCC, CONST-G, CONST-T and CONST-SJC, on each dataset AUD, CAD, JPY, NOK, SEK, EUR, GBP and NZD. The format is

Method Name

results for AUD
results for CAD
results for JPY
results for NOK
results for SEK
results for EUR
results for GBP
results for NZD

As in the manuscript, the results of the best method are shown in bold (with a "b" in front). The results of any other method are underlined (with a "u" in front) when the differences with respect to the best performing method are not statistically significant according to a paired t-test at \alpha = 0.05. The best technique is GPCC-T (the one with most "b"'s in front), followed by GPCC-G as discussed in the manuscript.

GPCC-G

0.1260
u 0.0562
b 0.1221
u 0.4106
u 0.4132
0.8842
u 0.2487
u 0.1045

GPCC-T

b 0.1319
b 0.0589
u 0.1201
b 0.4161
b 0.4192
b 0.8995
b 0.2514
b 0.1079

GPCC-SJC

0.1168
0.0469
0.1064
0.3941
0.3905
0.8287
0.2404
0.0921

HMM

0.1164
0.0478
0.1009
0.4069
0.3955
0.8700
0.2374
0.0926

TVC

0.1181
0.0524
0.1038
0.3930
0.3878
0.7855
0.2301
0.0974

DSJCC

0.0798
0.0259
0.0891
0.3994
0.3937
0.8335
0.2320
0.0560

CONST-G

0.0925
0.0398
0.0771
0.3413
0.3426
0.6803
0.2085
0.0745

CONST-T

0.1078
0.0463
0.0898
0.3765
0.3760
0.7732
0.2231
0.0875

CONST-SJC

0.1000
0.0425
0.0852
0.3536
0.3544
0.7113
0.2165
0.0796